# Brief communication: Tritium concentration and age of firn accumulation in an ice cave of Mt. Olympus (Greece)

Georgios Lazaridis[1,*], Konstantinos Stamoulis[2,*], Despoina Dora[1], Iraklis Kalogeropoulos[3], Konstantinos P. Trimmis[4]

[1]School of Geology, Aristotle University of Thessaloniki, Thessaloniki 54124, Greece
[2]Physics Department, University of Ioannina, Ioannina, 45221, Greece
[3]Hellenic Speleological Society, Thessaloniki, 54124, Greece
[4]Department of Anthropology and Archaeology, University of Bristol, Bristol, BS8 1TH, England
*These authors contributed equally to this work

*Correspondence to*: Georgios Lazaridis (geolaz@geo.auth.gr)

**Abstract.** Firn from an ice cave in the highest mountain of Greece, Mt. Olympus, was sampled and analysed to determine the tritium content in order to estimate rates of accumulation and to date the ice plug. The presence of a sharp tritium peak content indicating the nuclear testing era was expected to be preserved into ice beds. Tritium concentrations were found to vary from 0.9 to 11 TU. This peak did not appear in the analysed samples providing an upper age limit of less than 50 years for the oldest sampled layer. It is suggested that the rate of melting is responsible for the absence of older firn layers.

## 1 Introduction

Ice caves or caves hosting perennial ice accumulations (Persoiu and Lauritzen, 2018) in Greece are scattered throughout the country's latitude. In total, 76 records of ice caves in Greece have been processed according to Luetscher and Jeannin (2004) classification scheme and examined according to climatological criteria and particularly the prevailing air dynamics and glaciological characteristics, such as the type of ice, by Lazaridis et al. (2018). In Mt. Olympus, which is the highest mountain of the country, all the ice caves are classified as 'static with firn', where 'static' is interpreted as single entrance caves that form single down-sloping conduits. Ice of these caves, particularly the ones on the Eastern slopes of Olympus, has been exploited for years between the end of the 19th Century and the 1950s, to provide ice to villages and towns in the foothills of the mountain and across the southwestern Greek Macedonia.

In order to determine how old is the firn accumulation in these caves, measurements of the concentration of tritium in the ice were carried out at the Christaki Pothole in Mt. Olympus. Tritium is a hydrogen isotope that decays emitting beta particles of very low energy (Emax ~18.6 keV) with a half-life of $4500 \pm 8$ days (Lucas and Unterweger, 2000; Ehhalt et al., 2002). Tritium is produced in the upper atmosphere by the interaction of cosmic rays with the atoms of the atmosphere. This mechanism introduces tritium into the water cycle by the form of tritiated water and the normal concentration of tritium in precipitation is found to be 5 to 10 TU (Tritium Units, 1TU = 0.11919 Bq/L, Terzer-Wassmuth et al., 2022). During late '50s and early '60s nuclear era, hydrogen bomb detonations introduced huge amounts of tritium into the atmosphere, resulting in a sharp peak of tritium concentration in precipitation all over the world (Martell, 1963), especially in the northern hemisphere where tritium

concentrations were increased up to 6,000 TU, reached in 1963 (Cauquoin et al., 2016). In Greece, a maximum of 3,550 TU
was observed also in 1963. The tritium peaks in precipitation of the early 1960s, are expected to be preserved in ice sections,
if they have not melted since then. Previous studies have dated or provided constrains on the age of the ice deposits by using,
the presence (Kern et al., 2009; Borsato et al., 2004; Kern et al., 2018 ) or the absence (Kern et al., 2011) of the tritium peak.

## 2 Cave description and geological setting

The cave under investigation (Fig. 1; and supplementary information) is located on the NW slope of Mount Olympus
(N40.06898 E22.31350) at an altitude of 2,350 m, very close to the Christaki refuge. The sampling cave was firstly investigated
in 2016 and was included in the cadaster of ice caves in Greece, under the name Christaki Pothole (Lazaridis et al., 2018). The
cave was surveyed once every year, for three consecutive years (from 2016 to 2018), during summer or early autumn and ice
accumulation was detected in each survey. The entrance of the pit is about ten meters and continues with the floor inclining to
the west. Most of the cave is covered by an ice plug that overlies limestone gravels and small blocks.  The ice plug consists of
accumulated firn and snow. When surveyed, it was 1.5 m thick below the entrance and progressively thickening westwards,
reaching a maximal thickness of 4.5 m. The ice plug prevents any access to the westward continuation of the cave. The cave
is hosted in the Cretaceous crystalline limestones with dolomite interferences (Latsoudas and Sonis, 1985). The limestone
sequence has a total thickness of about 2,650 m and consists of limestones that gradually transitions to dolomite. While the
eastern slopes of Olympus are affected by the maritime air masses, the western slopes of Mount Olympus act as a barrier to
the hot and humid westerly air masses and thus experience enhanced orographic precipitation. Ice caves in Greece are directly
influenced by the prevailing climatic conditions (Perşoiu et al., 2021). The amount of precipitation in combination with the
high altitude, leads to snowfalls even during summer season (Sahsamanoglou, 1989). Snow accumulation has been measured
to reach over 2.5 m of depth within wide topographic depressions of Olympus, in altitude higher than 2,000 m (Styllas et al.,
2016).

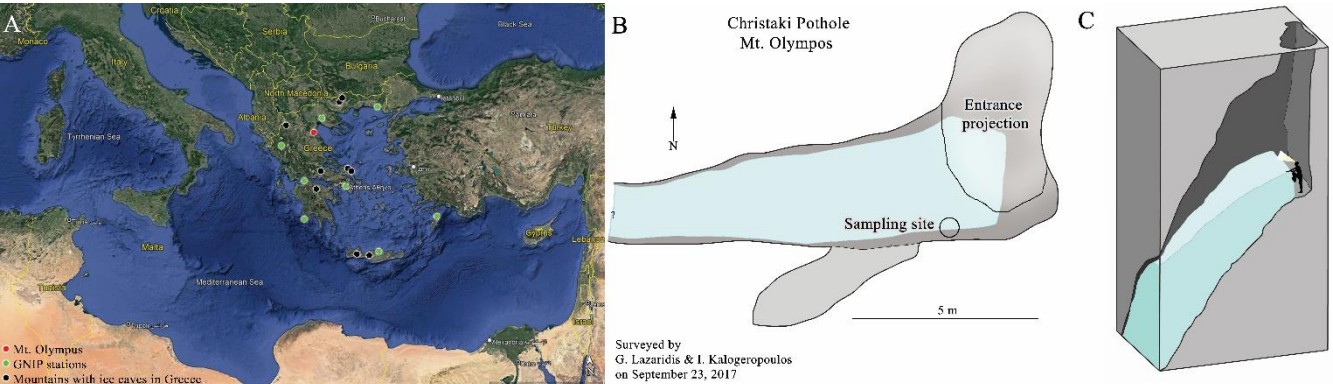

**Figure 1. A. Location of Mt. Olympus and the sampled cave Christaki Pothole; tritium monitoring stations (IAEA/WMO (2022).**
**The GNIP Database. https://nucleus.iaea.org/wiser); other mountains of Greece with ice caves (satellite image source from: ©**
**Google Earth 2022). B. Ground-plan with the sampling site depicted and 3D representation of the cave.**

## 3 Sampling and methods

On 23rd of September 2017, forty-one (41) ice samples were collected from the Christaki Pothole. Each sample was collected from a 2 m high section (Fig. 2), with a portable cup drill of 5cm diameter and about 2-3 cm depth, resulting in samples of 30 to 40 cm$^3$, at 5 cm intervals and spanning from the top of the ice section to the cave floor. Samples with odd numbering were selected for tritium content determination. Measurements were performed at the Archaeometry Center of the University of Ioannina.

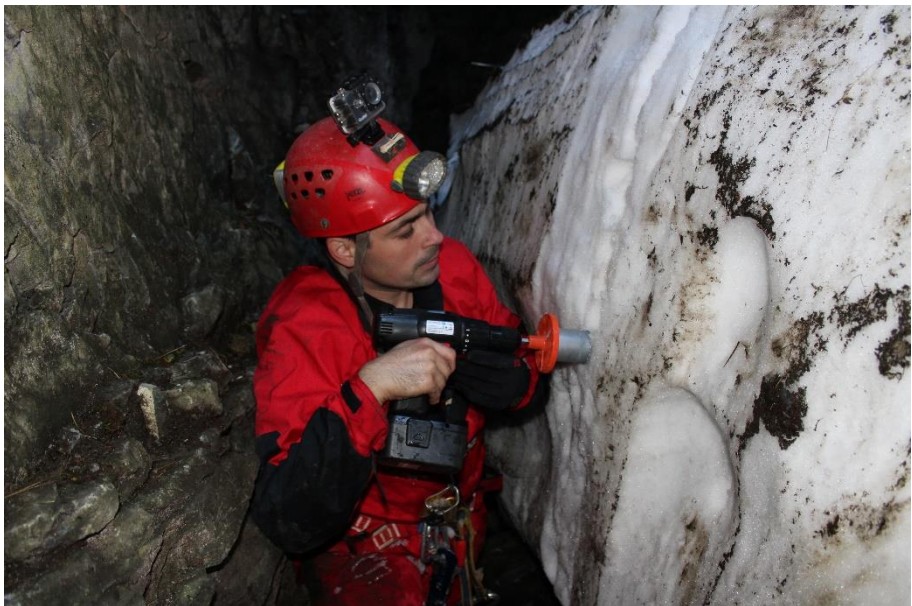

Figure 2. Sampling location and procedure, in the Christaki Pothole.

Tritium measurements were conducted in a Liquid Scintillation Analyzer (TR/SL 3700 Tricarb, Perkin Elmer). For each sample, 8 mL of melted ice, without electrolytic enrichment were added in a low-potassium borosilicate glass vial of 20 mL capacity and 12 mL of Ultima Gold LLT scintillation cocktail were added. The vial was closed and shaken to homogenize the solution and was measured for 1400 min typically. Background was also recorded at the same batch of samples. In order to establish the detection limit of the method, several background measurements were pooled and a mean value of $1.200 \pm 0.006$ cpm were selected as the representative value. The detection limit was calculated using the equation $DL = \frac{3 \cdot \sigma_B}{eff \cdot V \cdot 60 \cdot 0.11919}$ (TU), where $\sigma_B = 0.006$ cpm is the uncertainty for the background, eff = 0.26 the efficiency of the detector, V= 0.008 L the volume of the sample, 60 for min to sec and 0.119919 Bq/L=1 TU. The calculated value was DL=1.2 TU.

## 4 Results and discussion

The lack of well distinguished layers precluded the estimation of the age of the samples extracted from the ice deposit, by layer counting. However, during the sample processing, it was revealed that almost every ice sample contained remains of dust and soil which is the result of the surface debris deposited during the yearly cycle of snow accumulation during winter and partial melting during summer. Since the cave firn deposit was thought to have been accumulated for many decades, it was hypothesized that the above-mentioned atmospheric tritium peak would be found in the melted ice samples. Instead, tritium concentrations were found to vary only from 0.9 to 11 TU. This range of tritium concentrations could be attributed to different initial tritium concentrations in snowfall, either few years before the sampling year or even older. In comparison with tritium concentrations for selected years before the sampling year (2017), most of the tritium concentrations, corrected for decay, could result from precipitation up to 50 years before sampling year, which corresponds to the calendar year 1967, when the mean annual tritium content was in the range of 130-230 TU (Fig. 3, Table S2).

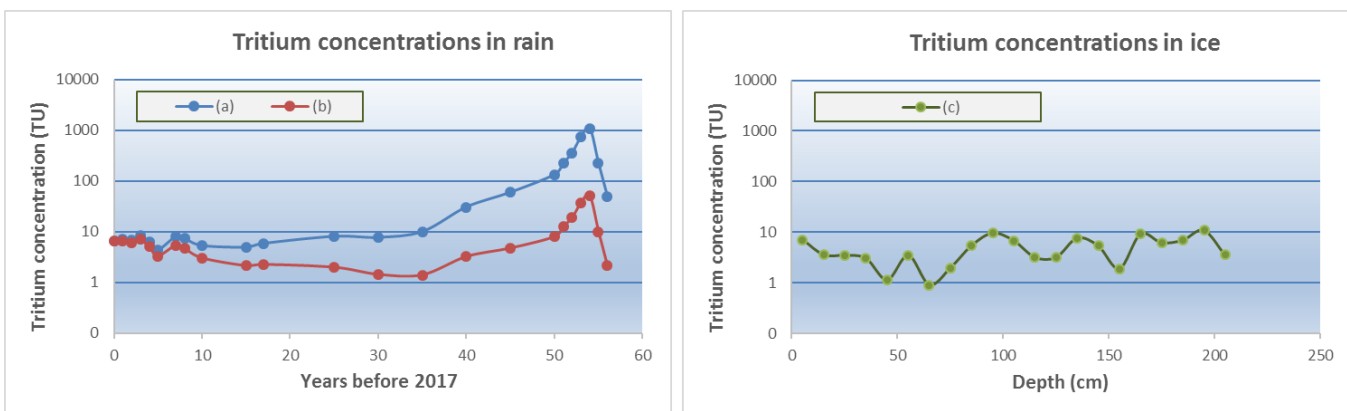

**Figure 3. Tritium concentration: (a) Annual mean values of tritium in rain samples from various monitoring stations in Greece (GNIP, IAEA, see Fig.1 for the sites of the stations and Table S3 for the coordinates and altitude). (b) Annual mean tritium values of line (a) corrected for decay at the year of ice sampling (2017). (c) Tritium content in ice samples, of odd numbering, with depth, from the Christaki pothole.**

Another consideration could be the case the deposited firn to be the result of the last decade precipitation when the annual variation of tritium concentration was in the same range. Then the local maximums observed in the tritium concentration through the samples, could be attributed to annual season maximums observed usually during the spring to summer months. This consideration could lead to seven to ten annual cycles observed in the preserved 2m high ice bed, concluding an annual accumulation rate to the firn of about 20-30 cm y$^{-1}$. Finally, the $^3$H activity levels, found in the ice samples, could not be attributed to precipitation fallen before the '50s , because in that case higher concentrations remaining from the high tritium concentrations during early '60s should have been preserved into some of the measured samples. This gives an upper limit of the possible age of the ice layers that were sampled during the campaign of September 2017. Considering this upper limit of 50 years for the base of ice deposit, the corresponding mean winter ice layer thickness is at least 4 cm y$^{-1}$; the result is in

accordance with the presence of debris in almost each sample, as well as with findings in other ice caves. In Monlesi Ice cave, Switzerland, an annual accumulation rate 7-11 cm $y^{-1}$ was estimated based on tritium and $^{210}$Pb measurements (Luetscher et al., 2007) and in Ledena Pit, a Croatian ice cave, an average accumulation rate of up to 12 cm $y^{-1}$ during period of 1963-1995 was also estimated (Kern et al., 2008). In other cases (Racine et al, 2022), the radiocarbon dating of organic remains in the

firn, reveal ages that span from modern times to almost 2700 BCE, depending the morphology, the microenvironment and the height of the accumulated firn.  For heights much larger than 2m of the case presented here, varying from 5 to 27 m above the base of the firn, the estimated mean annual accumulation rates were much lower, varying from few cm to less than 1 cm $y^{-1}$. However, the estimates of the present work, do not date the onset of cave glaciation, nor times at which the cave may have been completely ice free. The ice plug in ice caves melts when it is in contact with the bedrock (Bella and Zelinka, 2008;

Telbisz, 2019; Racine et al, 2022). Thus, the results are rather indicative of a melting rate that cannot support the preservation of ice that is older than 50 years. However, in deeper inaccessible parts of the cave older ice may be preserved.

## 5 Conclusions

The Christaki Pothole provided a two meters thick succession of ice samples that display relatively low concentration of tritium, without any pattern of an abrupt increase that could correspond to the tritium peak due to hydrogen bomb detonations

in early '60s. This indicates that accumulated firn beds were younger than fifty years in 2017. The absence of earlier beds is suggested to be the result of melting rate at the bottom of the ice plug or may occur in ice beds that are not accessible due to the morphology of the Christaki Pothole.

**Data availability**

All raw data are provided in the supplementary information.

**Competing interests**

The authors declare that they have no conflict of interest.

**Author contribution**

GL initiated the research; GL and KS organized field work and prepared the manuscript with inputs from all authors; KS

analyzed samples; GL and IK surveyed the cave and sampled the ice; GL, KS, IK did the field work; DD and KT collected information.

**Acknowledgments**

We are deeply grateful to Sakis Ntavlis and Vasilis Sidiropoulos, cavers of the Hellenic Speleological Society, Department of Northern Greece for their contribution during the field work and sampling. Tanguy Racine and Zoltán Kern are thanked for

their comments and suggestions, as reviewers, which improved the paper.

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
