# Peer review of "Brief communication: Tritium concentration and age of firn accumulation in an ice cave of Mt. Olympus (Greece)"

_The Cryosphere, 2022_

## Author Response (AR1)

**Brief communication: Tritium concentration and age of firn accumulation in an ice cave of Mt. Olympus (Greece) – Response to the referees**

Georgios Lazaridis[1,*], Konstantinos Stamoulis[2,*], Despina Dora[1], Iraklis Kalogeropoulos[3], Konstantinos P. Trimmis[4]

[1]School of Geology, Aristotle University of Thessaloniki, Thessaloniki 54124, Greece
[2]Physics Department, University of Ioannina, Ioannina, 45221, Greece
[3]Hellenic Speleological Society, Thessaloniki, 54124, Greece
[4]Department of Anthropology and Archaeology, University of Bristol, Bristol, BS8 1TH, England
*These authors contributed equally to this work

*Correspondence to*: Georgios Lazaridis (geolaz@geo.auth.gr)

**Response to referee #1**

General comments

This paper presents a new series of tritium measurements from firn sampled in an underground ice deposit of Mount Olympos. The lack of the so-called 'Tritium bomb peak' in the series (all current values are under 10 TU) is interpreted as evidence for a maximum age of ~50 years for the ice deposit of Christaki cave. I recommmend this communication for publication, provided some significant changes are made to the manuscript.

From the cave survey and the geometry of the ice body however, I believe there are limitations on the conclusions about the age of the ice deposit in this cave. It appears that there are indeed, according to the survey some deep ice layers that you could not access for sampling, which may have yielded the tritium peak. In this case, the maximum age of the ice could be older still. I stress that since the ~2 m sampled section was deposited in at most 50 years, the minimum rate of ice accumulation interpreted by the authors remains valid. I would encourage the authors to discuss the possibility of older ice being present, but inaccessible in the discussion, rather than the final line of the conclusion.

I feel this short communication is a relevant contribution as it provides evidence for a minimum mean annual mass balance in high elevation ice caves, which are still under researched objects of the cryosphere.

In general, the quality of the English could be significantly improved. A mix of British and American English was noted in the text (e.g., analysed in the Abstract, meters in the main body of text). Please see the technical comments for suggestions.

Specific comments

- I think the title should contain ice cave in the singular, because although there is a mention of neighbouring cave on mount Olympus, the age determination is only carried out at one of these sites. I say this because the geometry constraints of each cave may permit the accumulation of firn bodies with varying thicknesses and age within a very small karst area.

*We changed the title according to the reviewer's suggestion.*

- From figure S3, it looks like there are organic debris falling into the entrance shaft of the Christaki pothole. I think a way to continue/ corroborate this report would be radiocarbon dating any organic inclusions found in the ice. One would expect idea a fraction modern F14C > 1 from one such sample, given the hypothesis of 'young' firn in the caves of these areas.

40     *We thank the reviewer for this suggestion. It is among our future goals to use radiocarbon dating to access the age of ice accumulation in the caves of Mt. Olympus.*

- In several places, an 'ice column' is mentioned, which I assume is the section of ice that was sampled for tritium measurements. I would suggest changing this to 'ice section', because it might otherwise be confused with a pillar shaped congelation ice speleothem.

45     *We changed the word "column" to "section".*

- At the end of the introduction, I would also introduce previous ice cave studies which dealt with ice dating using the tritium peak, either present (1) or absent (2).

*The phrase "Previous studies have date ice deposits by using the tritium peak, which was either present (Kern et al., 2009) or absent (Kern et al., 2011)." is added at the end of the introduction.*

50
- At line 45, high average snow fall is mentioned - this could do with a citation and quantitative estimate of e.g., mean snow depth.

*By this sentence we wanted to emphasize on the high snowfall. Its depth can reach 3m in the area of the cave and the Christaki refuge. However, a reference is missing. So, we changed this phrase in order to use published information and provide some impression of the snowfalls in Mt. Olympus. "The amount of precipitation in*
55     *combination with the high altitude, leads to snowfalls even during summer season (Sahsamanoglou, 1989)."*

- Lines 60-66 - I find the line of reasoning hard to follow. You observe that every sample contains soil debris and dust, a material which was deposited onto the ice deposit during the summer period. You argue that ablation and overburden by new snow the following winter result in the dissemination of the material within the existing firn, thus obscuring the layers, making them indistinguishable. You then infer that the
60     maximum single annual layer thickness could not exceed that of the current ice deposit? If so, I do not see how it helps the following reasoning.

*We realized that this part of the text was hard to follow, and we agree with the reviewer. We deleted the sentence 64-66 that was not adding to the reasoning about the age of the ice body and we did some improvements to the lines 60-64.*

65
- I think sentence 64-66 could be deleted or made clearer in the text why it adds to the reasoning about the maximum age of the ice body.

*We agree with the reviewer and thus we deleted the phrase.*

- Lines 66-71 - this section of the text contains the main line of argument - i.e., corrected for decay, one would expect a > 130 TU tritium peak in an ice deposit whose age exceeds 50 years, provided there are no
70     hiatuses and provided the ice in the cave is not relict ice, currently melting. I think you have observations that show the snow surface is not ancient, but melts from the bottom. I have added some technical comments to have this section read better.

*We followed all the related technical comments suggested by the reviewer below.*

- Figure 1 - I think that this figure would benefit from a simplified overview map of the central
75     Mediterranean region (currently Figure S1), showing where exactly the site is in relation to the coastlines and maybe other known/published ice bearing sites from the Greek ice cave cadastre (this would serve to highlight and support the statement about how widespread ice caves are in Greece). I propose this because there is left-over width in the figure, so adding another small panel could help. Is the scale on Panel B the same as that of Panel A?

80    *Figure 1 is changed according to the reviewer's suggestion.*

Technical corrections

- l12: 'sharp raise of tritium' should read 'sharp rise of tritium' or 'sharp tritium peak'

- l14: It should probably be reworded to make clear here that the absence of the tritium peak provides an upper age limit for the deepest layer of the ice deposit.

85
- l18-19: 'and distribution': I am not sure what is meant here. The sentence could be reworded to make it clear that the classification scheme proposed by Luetscher and Jeannin (2004) helps differentiate ice caves based on their morphology (hence ventilation pattern) and the type of ice they contain (primarily firn, or congelation ice).

- l22: 'have been exploited for years...' should read 'has been exploited for years between the end of the 19th
90    Century and the 1950s'

- l23: 'to villages and and town at the foothills' should read 'to villages and towns in the foothills'

- l25 'were applied at' should read 'were carried out at'

- l26-28: this pair of sentences would benefit from citations, eg: (3) (4)

- l30 'resulting to a sharp peak' should read 'resulting in a sharp peak', also could use citation (5)

95
- l31-32: could be better formulated such as: 'up to 6,000 TU in Canada and Austria, reached in 1953 and 1961, respectively (Cauquoin et al., 2016).'

- l33-34: could be reformulated: 'The tritium peaks in precipitation of the early 1960s ...'

- l35: 'on the NW slope of the mountain' could read 'on the NW slope of mount Olympos'

- l36: 'the list of ice caves in Greece' could be reformulated as 'the Greek ice cave cadaster'

100
- l37: replace 'first descent' with 'entrance pit'

- l40: 'progressively goes thicker to the west, reaching about 4.5 m of thickness' could read 'progressively thickening westwards, reaching a maximal thickness of 4.5 m'

- l40-41: again maybe reformulate to: 'The ice plug prevents any access to the westward continuation of the cave'.

105
- l42: 'is consisted of' should either read 'is composed of' OR 'consists of'

- l43 'gradually reduce their composition to dolomite' should read 'gradually transition to dolomite'

- l43-45: reformulating the sentence so it could read 'The western slopes of Mount Olympos act as a barrier to the hot and humid westerly air masses and thus experience enhanced orographic precipitation'.

- l52 'from a section of 2 m high' should read 'from a 2 m high section (Figure 1A)'

- l53: 'in 5cm intervals' should read 'at 5 cm intervals'

- l54: 'chosen to be measured for tritium' could read 'selected for tritium content determination'

- l55: could reformulate to have no number at the start of the sentence? e.g., 'For each sample, 8 mL of melted ice were ...'

- l60: I think this sentence could be reformulated to highlight the fact the lack of layers precluded the estimate of the age of the ice deposit by layer counting.

- l64: I don't think incineration is the correct word: 'dissemination' perhaps?

- l66-67 I would reformulate the sentence to 'Since the cave firn deposit was thought to have accumulated for many decades, it was hypothesised that the above-mentioned atmospheric tritium peak would be found in the melted ice samples'.

- l70: 'could be resulted from' should read 'could result from'
- l71: 'annual mean tritium' should read 'mean annual tritium'

- l77: The sentence could be reformulated thus: 'Considering this upper limit of 50 years for the base of ice deposit, the corresponding mean winter snow layer thickness is at least 4 cm y-1'.

- l81: capitalise 'Croatian ice cave'.

- l83: Sentence could be reformulated thus: 'However, these estimates do not date the onset of cave glaciation, nor times at which the cave may have been completely ice free'.

*All the above technical corrections have been followed (lines 83-126).*

- l84: could you provide a reference for this statement?

*Actually, this was an indirect assumption that is provided to support the possibility of ice melting when it comes in contact with the bedrock. We changed the phrase, and we added two references that support this statement.*

Suggested References:

- (1) Kern et al, 2009 High-resolution, well-preserved tritium record in the ice of Bortig Ice Cave, Bihor Mountains, Romania

- (2) Kern et al, 2011, Isotope hydrological studies of the perennial ice deposit of Saarhalle, Mammuthöhle, Dachstein Mts, Austria

- (3) L.L. Lucas, M.P. Unterweger Comprehensive review and critical evaluation of the half-life of tritium J. Res. Natl. Inst. Stand. Technol., 105 (4) (2000), pp. 541-549

- (4) D.H. Ehhalt, F. Rohrer, S. Schauffler, W. Pollock Tritiated water vapor in the stratosphere: vertical profiles and residence time J. Geophys. Res., 107 (D24) (2002), p. 4757, 10.1029/2001JD001343

- (5) E.A. Martell On the inventory of artificial tritium and its occurrence in atmospheric methane J. Geophys. Res., 68 (1963), pp. 3759-3769

*All the above references were very useful and were added to the manuscript.*

**Response to referee #2**

1. As a basic requirement site photos and/or sketched stratigraphic profile should be mandatory to understand the sampling strategy and see the visual occurrence of the deposit and the sampling spots. In lack of such evidence a statement like "indistinguishable ice layers and thus it was impossible to make a direct estimation of the age of the ice in the column." " is unsubstantiated.

*Due to the ablation of the firn (late September sampling period) there was no clear evidence of the stratigraphy as can be derived from the provided photo. So, the sampling strategy was to take samples in a subsequent way, also as seen in the provided photo, although not clearly.*

2. Methodological description needs some more details. E.g., Did you apply electrolytic enrichment? If yes please give some details, if no please mention that. What was the critical limit and/or detection limit?

*We did not perform electrolytic enrichment because the Laboratory does not have well established curves of tritium enrichment efficiencies (line 68). The detection limit is added in the text (Detection Limit =3sqrt(B)/(T eff V a) where B=backround counts, T= measurement time (min), eff=efficiency of the detector for Tritium=0.25, V=8mL, a=0.11919 Bq/L/TU) DL=1.2 TU. All mentioned in lines71-75 in revised manuscript.*

3. The caption of Fig 2 says that the annual mean tritium values of various Greek stations are used as reference. It is not a bad approach however I think it would be necessary to show the location of the considered stations in a map. (By the way, Fig 1 should be completed with an additional panel showing the location of the cave, so the nearest GNIP stations can be marked in this map.)

*Table with the stations GNIP in Greece. Short introduction is added in the caption of Fig. 2 and this table in the supplement.*

| Site | Latitude | Longitude | Altitude |
|---|---|---|---|
| *Alexadroupolis* | *40.849998* | *25.879999* | *6* |
| *Athens* | *37.900002* | *23.73* | *27* |
| *Heraklion* | *35.330002* | *25.18* | *47* |
| *Methoni* | *36.830002* | *21.719999* | *33* |
| *Patras* | *38.279999* | *21.790001* | *100* |
| *Rhodes* | *36.380001* | *28.1* | *42* |
| *Thessaloniki* | *40.669998* | *22.959999* | *32* |
| *Ioannina (non GNIP)* | *39.663611* | *20.852222* | *480* |
| *Christaki pothole* | *40.068954* | *22.313373* | *2290* |

4. However, I suggest considering the prediction from the recently released study (Terzer-Wassmuth et al., 2022) as a reference or as a continuous interpolated product covering the 1950 to 2010 period (Jasechko&Taylor 2015) could be used.
   a. Terzer-Wassmuth, S., Araguás-Araguás, L.J., Copia, L. et al. High spatial resolution prediction of tritium (3H) in contemporary global precipitation. Sci Rep 12, 10271 (2022). https://doi.org/10.1038/s41598-022-14227-5
   b. Jasechko, S., & Taylor, R. G. (2015). Intensive rainfall recharges tropical groundwaters. Environmental Research Letters, 10(12), 124015

*Studying the Terzer-Wassmuth et al. (2022) paper we tried to predict tritium values in the investigation area using the models provided but the calculated values were obviously very high maybe due to the misunderstanding of the parameters and their values that must be used for the site. In any case the map provided for the area of Greece give values of tritium concentrations in the range we measured in our Laboratory.*

5. The authors explain why the studied ice samples could not represent accumulation from the so-called bomb-peak period. However, I think, it should be also explained in a sentence or in a brief section how they can exclude pre-1950 origin.

*The samples were collected from the top of the firn to a depth of 2 meters in a back-to-back way. Thus, the samples from the top are from the previous years. If there were ice residues from years before 50s then we should measure at some samples high tritium concentrations from the 60s as mentioned in detail into the manuscript. Short discussion is added into the manuscript (lines 98-99).*

**Technical revisions**

line 13: I think "indicating" would be a more suitable word here instead of "because". In addition, the range of the measured 3H activities could be mentioned in the abstract.

*Followed in lines 13-14*

line 18: I suggest citing the chapter (Pennos et al., 2018) of the Ice Cave Book here. *Pennos, C., Styllas, M., Sotiriadis, Y., and Vaxevanopoulos, M.: Ice caves in Greece, in: Ice caves, edited by: Persoiu, A. and Lauritzen, S. E., Elsevier, Amsterdam, the Netherlands, 385-397,* https://doi.org/10.1016/B978-0-12-811739-2.00018-8 2018.

*The research that we mention (Lazaridis et al., 2018) contains data from 76 caves, the one that is suggested here contains only three of the already included in the mentioned research caves. We thank the referee for the comment, but we don't find useful and necessary the suggested citation.*

line23: I think a supporting reference for this statement is needed.

*Unfortunately, there is not a reference for this statement. The information comes from interviews with local people. However, we find it very interesting and useful to publish*.

line26: I think Lucas&Unterweger 2000 should be cited after the half-life of tritium. *Lucas, L.L. and Unterweger, M.P. 2000: Comprehensive review and critical evaluation of the halflife of tritium. Journal of Research of the National Institute of Standards, Technology 105, 541–49.*

*The suggested reference was added and the Ehhalt et al. 2002 as well.*

lines31-32: I think the end of this sentence seems to be a fragment which can be deleted.

*The sentence has already been modified due to a comment form the first referee.*

210

line42-43: Why these info (e.g., total thickness of limestone sequence, dolomitic composition) is useful for this study?

*We believe that is very useful to describe the setting of the cave and the geology of the area and we consider it as necessary information.*

215

line 44: I suspect you should replace "gas" with "air".

*The sentence has already been modified due to a comment form the first referee.*

line46: Please give numerical expression for "high average snowfall".

220 *The sentence has already been modified due to a comment form the first referee.*

lines6 60-61: Unclear sentence. Did you mean that ice layers (or any stratigraphic units) were indistinguishable in the sampled ice column?

*The sentence has already been modified due to a comment form the first referee.*

225

line 68: I suggest replacing "0.9-11" with "0.9 to 11".

*The suggestion was followed.*

line81: I think 210Pb should be written instead of "radon". In addition, please, capitalize Croatia in the same line.

230 *Replaced and correctly spelled line 105*

Finally, I think a recent TC paper (https://doi.org/10.5194/tc-15-2383-2021) should be considered in an extended discussion since similar deposits were considered also in that study.

235 *We followed the suggestion, we consulted the paper and added to the references.*

In an ultimate comment I'd like to refer to the other review. I completely agree with the comments and suggestions of Dr Tanguy Racine. A related suggestion is that beside 2011 paper about Mammuthöhle ice cave I suggest that a more recent one (DOI:10.1017/RDC.2018.96) could be a more useful reference for the revision.

*The suggestion was followed as we find the research relevant and necessary to refer to.*

---

## Author Response (AR2)

**Brief communication: Tritium concentration and age of firn accumulation in an ice cave of Mt. Olympus (Greece) – Response to the referees**

Georgios Lazaridis[1,*], Konstantinos Stamoulis[2,*], Despoina Dora[1], Iraklis Kalogeropoulos[3], Konstantinos P. Trimmis[4]

[1]School of Geology, Aristotle University of Thessaloniki, Thessaloniki 54124, Greece
[2]Physics Department, University of Ioannina, Ioannina, 45221, Greece
[3]Hellenic Speleological Society, Thessaloniki, 54124, Greece
[4]Department of Anthropology and Archaeology, University of Bristol, Bristol, BS8 1TH, England
*These authors contributed equally to this work

*Correspondence to*: Georgios Lazaridis (geolaz@geo.auth.gr)

**Response to the remark of the editorial team**

**We changed the caption of figure 1 in order to include the copyright statement for the map used.**

**Figure 1. A. Location of Mt. Olympus and the sampled cave Christaki Pothole; tritium monitoring stations (GNIP); other mountains of Greece with ice caves (satellite image source from: Google Earth 2022). B. Ground-plan with the sampling site depicted and 3D representation of the cave.**

We also changed the name of Despina Dora to Despoina Dora.

---

## Author Response (AR3)

Point to point response to the reviewers

1st reviewer

1.  **l35 I would rephrase this slightly to give nuance about the fact that ice can be dated when the peak is present, whereas its age can be constrained in the absence of the tritium peak. I would suggest something like: Previous studies have dated or provided constraints on the age of ice deposits by using the presence (Kern et al, 2009) or absence (Kern et al, 2011) of the tritium peak.**

We rephrased the line according to the suggestion of the reviewer and we added two more references (Borsato et al., 2004; Kern et al., 2018) for the case of the presence of the tritium peak.

2.  **l46: change 'transits to dolomite' to 'transitions to dolomite**

We changed the phrase according to the suggestion

2nd reviewer

Regarding the major comments:

1.  **Site description needs crucial details and more climate information should be also necessary. e.g.: When was the cave discovered? or 2017 is the date of discovery? Can you provide evidence that the firn deposit sampled in 2017 is not an exceptional seasonal phenomenon at this site? (e.g.: did you observed firn in this cave before or after 2017?) Quantitative data (at least estimates) about the annual precipitation amount and the typical snow depth should be provided.**

The year that the cave was firstly investigated was added (lines 40-41, in the revised manuscript). We support the opinion that the firn deposit sampled in 2017 is not an exceptional seasonal phenomenon, based on our observations before and after the sampling year (lines 42-43, in the revised manuscript). A phrase that describes the snow depth and a reference were added (lines 53-54, in the revised manuscript).

2.  **The atmospheric reference data in Fig2A should be updated to 2017 and it should be discussed whether the studied ~2m firn represent a few years of the past ~10yrs. Beside the annual mean tritium values mean 3H activities of the snow season could be also considered as a more suitable benchmark. A related question is that if the ~2m profile cover only few years than the fluctuation between ~1 TU and ~10 TU might reflect some seasonal cycle? It should also be discussed. If this is viable than the inferred accumulation could be ~70cm/yr bringing substantial changes to the discussion.**

We updated the data with our tritium in precipitation measurements of the period suggested (2007-2017 with a gap during 2011) since there are no data at the GNIP database for the same period and for stations in Greece. Also, we added a brief discussion (lines 89-93, in the revised manuscript) for the case the data should represent accumulated ice from the last decade. This addition does not change the whole idea of

the discussion since we suppose, from the data collected, that the accumulated ice is not older than 50 years from the year of the sample collection and thus the minimum annual accumulation rate is about 4 cm $y^{-1}$. So, the rates mentioned in the newly added discussion of about 20-30 cm $y^{-1}$ are in consistency with our argument.

3. ***The discussion could be also improved. Authors should consider the observations from a recent study of one of the previous reviewers: https://doi.org/10.1038/s41598-022-15516-9 In addition, I suggest carefully considering the differences between ice deposits from firn and congelation ice. (The usual accumulation of congelation ice is lower by orders of magnitude compared to firn deposits!). It would be interesting, especially the TC readers what is the situation of this firn deposit following the extreme weather of 2018-2019 reported in another study (https://tc.copernicus.org/articles/15/2383/2021/ ) in this journal.***

We added a phrase mentioning the results of the proposed recently published work of Racine et al, 2022, stressing that there are ice caves where the preserved firn may have age up to 4700 years BP (~2700 BCE). Regarding the possibility of updating the situation of the cave at Mt Olympos through the recent years 2018-2019, it is impossible to do so because there wasn't organized any sampling campaign in the cave the recent years.

Regarding the specific comments:

4. ***line 30: I suggest writing '5 to 10 TU' instead of '5-10 TU'***

We followed the suggestion

5. ***line 33: I suggest removing „in Canada and Austria" since northern hemisphere is already mentioned in the previous line in the same sentence.***

We removed it

6. ***line 36: Instead of citing the study from 2011 please consider the follow up study (DOI:10.1017/RDC.2018.96 ) which was able to detect 3H activities corresponding to the „bomb peak" period from the same cave ice core. In addition, 3H activities mirroring the concentration changes of atmospheric precipitation have been reported from a 2.5 m topmost part of Grotta del Castelletto di Mezzo ice deposit (Brenta Dolomites, N Italy) as well. This paper (url: https://www.academia.edu/download/39282184/0deec528405fbec31f000000.pdf ) should be also mentioned here.***

Following the suggestions the lines were rephrased as follows: "*Previous studies have dated or provided constrains on the age of the ice deposits by using, the presence (Kern et al., 2009; Borsato et al., 2004; Kern et al., 2018 ) or the absence (Kern et al., 2011) of the tritium peak.*"

7. **lines 62-63: Please change 'ml' to 'mL' (Litre was abbreviated by capital letter at the other occurrences as well.)**

Changed.

8. **lines 76-78: I think I understood what you were saying, but the sentence is quite long and convoluted. Please consider splitting it up to simple statements.**

We rephrased the sentence as follows: "*Instead, tritium concentrations were found to vary only from 0.9 to 11 TU. This range of tritium concentrations could be attributed to different initial tritium concentrations in snowfall, either few years before the sampling year or even older.*"

9. **line 85: The sentence sounds strange. The measurements obviously cannot be attributed to anything before the '50s since those were performed after September of 2017. If I can follow the logic the Authors wish to argue that the 3H activity levels found in the ice samples could not be attributed to precipitation fallen before the 1950s. Please consider rephrasing.**

Rephrased as follows: "*Finally, the $^3H$ activity levels, found in the ice samples, could not be attributed to precipitation fallen before the '50s, because in that case higher concentrations remaining from the high tritium concentrations during early '60s should have been preserved into some of the measured samples.*"

Finally, regarding Figures:

10. **Figure 1. Please cite the GNIP database here in the caption. Please follow to the instruction of the WISER page: IAEA/WMO (current Year). Global Network of Isotopes in Precipitation. The GNIP Database. Accessible at:** *https://nucleus.iaea.org/wiser*

The capture was changed: "…tritium monitoring stations (IAEA/WMO (2022). The GNIP Database. https://nucleus.iaea.org/wiser); other mountains…"

11. **Figure S4: I suggest moving this field photo from the supplement to the main document. This give an impression about the studied firn deposit.**

The figure S4 was moved to the main document as suggested (Figure 2, in the revised manuscript).

---

## Author Response (AR4)

Point to point response to the comments of minor revision

We corrected the numbers in figures and typos in text.